# Does laryngeal reinnervation or type I thyroplasty give better voice results for patients with unilateral vocal fold paralysis (VOCALIST): study protocol for a feasibility randomised controlled trial

Helen Blackshaw,[1] Paul Carding,[2] Marcus Jepson,[3] Marina Mat Baki,[4] Gareth Ambler,[5] Anne Schilder,[1] Stephen Morris,[6] Aneeka Degun,[1] Rosamund Yu,[7] Samantha Husbands,[3] Helen Knowles,[8] Chloe Walton,[2] Yakubu Karagama,[9] Kate Heathcote,[10] Martin Birchall[1]

For numbered affiliations see end of article.

**Correspondence to**
Dr. Helen Blackshaw;
h.blackshaw@ucl.ac.uk

## ABSTRACT

**Introduction** A functioning voice is essential for normal human communication. A good voice requires two moving vocal folds; if one fold is paralysed (unilateral vocal fold paralysis (UVFP)) people suffer from a breathy, weak voice that tires easily and is unable to function normally. UVFP can also result in choking and breathlessness. Current treatment for adults with UVFP is speech therapy to stimulate recovery of vocal fold (VF) motion or function and/or injection of the paralysed VF with a material to move it into a more favourable position for the functioning VF to close against. When these therapies are unsuccessful, or only provide temporary relief, surgery is offered. Two available surgical techniques are: (1) surgical medialisation; placing an implant near the paralysed VF to move it to the middle (thyroplasty) and/or repositioning the cartilage (arytenoid adduction) or (2) restoring the nerve supply to the VF (laryngeal reinnervation). Currently there is limited evidence to determine which surgery should be offered to adults with UVFP.

**Methods and analysis** A feasibility study to test the practicality of running a multicentre, randomised clinical trial of surgery for UVFP, including: (1) a qualitative study to understand the recruitment process and how it operates in clinical centres and (2) a small randomised trial of 30 participants recruited at 3 UK sites comparing non-selective laryngeal reinnervation to type I thyroplasty. Participants will be followed up for 12 months. The primary outcome focuses on recruitment and retention, with secondary outcomes covering voice, swallowing and quality of life.

**Ethics and dissemination** Ethical approval was received from National Research Ethics Service—Committee Bromley (reference 11/LO/0583). In addition to dissemination of results through presentation and publication of peer-reviewed articles, results will be shared with key clinician and patient groups required to develop the future large-scale randomised controlled trial.

**Trial registration number** ISRCTN90201732; 16 December 2015.

## INTRODUCTION

The larynx provides the main tool for human social interaction. The vocal folds are innervated by the recurrent laryngeal nerve (RLN) and when this nerve supply to one vocal fold is interrupted, unilateral vocal fold paralysis (UVFP) ensues. Patients suffering with UVFP typically display a weak, hoarse voice[1] that tires easily, with some also suffering from breathing and swallowing problems. Quality of life of these patients is significantly reduced, with UVFP affecting their ability to interact socially and perform job functions.[2][3]Therefore, optimum restoration of the paralysed larynx is important for patients to lead full personal and professional lives.

The aim of treatment for UVFP is to achieve optimum vocal fold (VF) closure for voice production. Where speech therapy alone does not improve the voice sufficiently, surgical treatment is undertaken. While short-term relief of symptoms can be achieved via injection of the paralysed VF with a material to improve its position for the functioning VF to close against it, permanent results are only possible by two types of operation: surgical medialisation and laryngeal reinnervation. One method of the former is to insert a synthetic implant to move the paralysed VF into a better position for speaking (type1 thyroplasty), while the latter resupplies nerves to improve muscle

**BMJ**

## Strengths and limitations of this study

► This is the first clinical trial to be run in the UK to compare the effectiveness of two surgical approaches for restoration of laryngeal function in adults with unilateral vocal fold paralysis (UVFP).

► This study has been developed in response to: (1) patient demand and (2) a systematic review that reported a low quality of current literature available on laryngeal reinnervation and proposed that a formal prospective trial was required using standardised, internationally accepted outcomes.

► A similar trial, comparing laryngeal reinnervation and type I thyroplasty in adults with UVFP, was previously run in the USA, but was suspended prematurely due to irregularity in obtaining inform consent and low accrual. We acknowledge the challenges associated with recruiting to a phase III surgical study of this type and have sought to overcome them by: (1) learning from this previous trial; (2) designing this rigorous feasibility study to be conducted prior to running a larger scale phase III randomised controlled trial and (3) involving patients with UVFP throughout the feasibility study design, set up, analysis and dissemination processes.

► Our research team is multidisciplinary and our study design incorporates both qualitative and quantitative components.

► While our study focuses on voice-related criteria for inclusion, our design includes outcomes relating to both the voice and swallowing.

► Although participants and study personnel are not blinded to their allocated treatment group, the speech therapist and neurophysiologist outcome raters are blinded.

► Very few ear, nose and throat surgeons in the UK are trained to perform laryngeal reinnervation and so our study is limited to recruiting at those relevant UK sites.

► While we aim to follow-up all participants for 12 months, budget and time constraints may limit some of the latterly recruited participant follow-up to 6 months. However, all patients are followed up clinically in the long term.

bulk and tone (non-selective laryngeal reinnervation). Type I thyroplasty is a 1-hour operation under local anaesthetic where the medium grade silastic implant is inserted through a window in the cartilage lamina of the larynx. This implant medialises the paralysed VF in the midline position enabling the normal opposite vocal fold to make a firmer contact, producing a stronger voice. However, voice improvement may be short-term due to progressive atrophy of the denervated laryngeal muscles.[4] Furthermore, this procedure does not restore vocal fold tension, meaning that pitch variation remains suboptimal.[5 6] Reports on short and long-term voice outcomes of patients with thyroplasty are sparse and methodologically poor.[7] The newer approach to restoring laryngeal function in UVFP patients, non-selective laryngeal reinnervation, is a 2-hour operation under general anaesthesia in which the injured RLN and the donor nerve are identified and anastomosed. Its aim is to restore innervation to the larynx by 'borrowing' the activity of other motor nerves, specifically the ansa cervicalis. As such it may re-establish the tone and bulk of the vocal folds during speech and re-enable pitch control, resulting in normal or sufficient voice. The operation is safe and results in sustained improvement of glottal closure, maintenance of the vocal fold edge

and significant improvements in voice quality have been reported.[8]

A systematic review of studies on laryngeal reinnervation revealed that the quality of current literature is low and a formal prospective trial was recommended using standardised, internationally accepted outcomes.[9] Before embarking on a trial that will use public resources and the time of patients and healthcare staff, we need to confirm that running a trial is feasible and that the chosen outcome measures truly reflect voice quality and patients' perceptions.[10 11] For these reasons, and others given above, a feasibility study is critical prior to designing, with patients, a multicentre, phase III randomised clinical trial of treatment for this disabling condition.

## METHODS AND ANALYSIS
### Objectives
The primary research objective is to determine the feasibility of performing a multicentre phase III randomised controlled trial (RCT) of laryngeal reinnervation versus type I thyroplasty for adult patients with UVFP.

The secondary research objectives are to test the feasibility of:

1. The multicentre recruitment process, including continuous improvement to the process based on qualitative methodology;
2. Recruiters being able to present true equipoise with the treatment arms;
3. The randomisation processes and to investigate reasons for any difficulties that affect recruitment;
4. The use of the following characteristics of the proposed primary and secondary outcomes: variability across patients, variability over time, differences in outcome between randomised groups over time;
5. Process of follow-up visits;
6. Means of gathering health economics and health-related quality of life data suitable for measuring cost-effectiveness.

### Study design
The feasibility study will consist of two research approaches run in parallel:

1. Qualitative: the Quintet Recruitment Intervention (QRI) to understand the recruitment process and how it operates in clinical centres, so that sources of recruitment difficulties can be identified and suggestions made to change aspects of design, conduct, organisation or training that could then lead to improvements in recruitment[12];
2. Quantitative: a small randomised surgical trial of 30 patients to compare non-selective laryngeal reinnervation to type I thyroplasty. Patients will be randomised in a 1:1 ratio.

The study is single blinded whereby the raters of the secondary quantitative outcome of acoustic measures will be blinded to the time-point of the data recordings (preoperative or postoperative) and blinded to

surgical intervention (laryngeal reinnervation or type I thyroplasty).

## Participants

### Inclusion criteria

1. UVFP due to unilateral recurrent laryngeal nerve paralysis of traumatic, iatrogenic or idiopathic origin of between 6 and 60 months duration or symptoms that have not sufficiently improved with speech therapy alone, as determined by the patient and agreed by a multidisciplinary clinical team, after 6 months and pending a surgical decision;
2. Age from 18 to 70 years old;
3. Male or female;
4. Able to provide informed consent;
5. A significant voice disorder as measured by perceptual rating (GRBAS (grade, roughness, breathiness, asthenia, strain) Grade≥2)[13] and Voice Handicap Index (VHI-10 score >16)[14];
6. Common laryngeal electromyography (LEMG, neurophysiological) criteria (Koufman Grades 2–5)[15] in either the thyroarytenoid or posterior cricoarytenoid muscle on the paralysed side.

### Exclusion criteria

1. Impaired vocal fold mobility but a normal EMG (Koufman Grade I);
2. Severe lung disorders;
3. Structural vocal fold lesions such as polyp;
4. Previous laryngeal framework surgery;
5. Cricoarytenoid joint fixation;
6. Significant non-laryngeal speech abnormality (severe dysarthria determined by a panel of trained speech therapists);
7. Previous level 2, 3 or 4 thyroid neck dissection;
8. Previous ipsilateral surgical neck dissection;
9. Previous radiotherapy to the head and neck;
10. Laryngeal injection of a rapidly absorbable material in the last 6 months;
11. Previous laryngeal injection of a non-rapidly absorbable material (eg, bioplastics, voice examiner);
12. Neuromuscular disease affecting the larynx or multiple cranial nerve palsies.

## Setting

The study is sponsored by University College London Hospitals National Health Service (NHS) Foundation Trust. Potential participants will be recruited from three UK hospitals; the Royal National Throat, Nose and Ear Hospital (University College London Hospitals NHS Foundation Trust), Manchester Royal Infirmary (Central Manchester University Hospitals NHS Foundation Trust) and Poole Hospital (Poole Hospital NHS Foundation Trust). The study is run through the University College London evidENT team, with qualitative data analysed by the University of Bristol and quantitative data analysed by University College London and Australian Catholic University.

## Interventions

Laryngeal reinnervation: the surgical technique is an unselective reinnervation[16] that aims to improve the muscle tone and bulk. Performed under general anaesthesia, the injured RLN and the donor nerve are identified and anastomosed.

Type 1 thyroplasty: this medialisation/augmentation technique is a static technique,[17] performed under local anaesthesia that aims to improve the positioning of the paralysed vocal fold. It uses a medium grade silastic implant, cut by the surgeon into the correct size for the patient. This size is determined intraoperatively by using a measuring device while listening and visualising the larynx with flexible fibreoptic scope simultaneously. Concurrent arytenoid adduction is permitted to be performed in cases where there is a significant posterior gap that cannot be approximated with the silicon block alone.

## Outcome measures

The primary outcomes are from those patients who were eligible to enter the study:

1. Whether the patient was randomised (yes/no);
2. Whether the participant successfully received the allocated operation (yes/no);
3. Whether the participant completed the trial (yes/no).

The quantitative secondary outcomes are:

1. General health status measured using the validated EuroQoL 5 Dimensions 5 Levels[18] questionnaire. Participants will self-administer the questionnaires at baseline, 6 and 12 months.
2. Voice-related quality of life measured using the validated VHI-10.[14] Participants will self-administer the questionnaires at baseline, 6 and 12 months.
3. Swallowing-related quality of life measured using the validated Eating Assessment Tool (EAT-10.[19] Participants will self-administer the questionnaires at baseline, 6 and 12 months.
4. Vocal fold vibration measured detected using laryngeal videostroboscopy and graded using the Stroboscopy Research Instrument scale.[20] Data will be collected at baseline, 6 and 12 months, anonymised and analysed by three independent trained raters.
5. Perceptual voice quality measured using the GRBAS[13] four-point grading scale (0—normal or absence of deviance; 1—slight deviance; 2—moderate deviance and 3—severe deviance) to evaluate 'G'; the grade of the overall voice quality, 'R'; roughness or harshness, 'B'; breathiness, 'A'; asthenia and 'S'; strain. Voice recordings will be collected at baseline, 6 months and 12 months, anonymised and graded by three independent experienced speech therapists trained to enhance the intra-rater and inter-rater agreement.[21]
6. Acoustic voice quality of the participants' voice recording will be captured and measured using the 'On person Rapid Voice Examiner',[22] a specially designed portable voice analysis application that

will be loaded onto Apple iPod Touch devices at each recruitment site. The noise-to-harmonic ratio, harmonic-to-noise, jitter, shimmer, shimmer dB, fundamental frequency, pitch range and maximum phonation time (MPT) data will be collected at baseline, 6 months and 12 months.

7. Swallow volume, capacity and speed measured using the validated 100 mL water Swallow Test[20] at baseline, 6 months and 12 months.

8. Laryngeal muscle activity detected using LEMG and graded using the Koufman grading scale.[15] Data obtained will be collected at baseline and 12 months, anonymised and analysed by three independent neurophysiologists.

9. Health economic cost analysis of laryngeal reinnervation and thyroplasty procedures including the costs of the index procedure (staff costs, theatre costs, costs of consumables, recovery costs).

10. Feasibility assessment of conducting a full economic evaluation of laryngeal reinnervation versus thyroplasty alongside the future large-scale RCT. Data will be collected to identify: (1) the main NHS/Personal Social Services cost components; (2) the main costs borne by patients and families and (3) the resource use and unit cost data required for each of these cost components and how best to source these.

The qualitative secondary outcomes are:

1. Whether members of the Study Management Group (SMG) and recruiters are in equipoise with the treatment arms;
2. Patient views and beliefs on randomisation, recruitment and retention processes;
3. Patient and surgeon/speech therapist views on chosen outcome measures;
4. Recruiter–patient interactions.

### Sample size

The sample size of 30 participants (15 per arm) will allow us to demonstrate the feasibility of conducting a full-scale randomised trial in the future. This study is not formally powered to detect differences (although we note that the sample would be able to detect very large effects, for example, a standardised effect of 1 with approximately 80% power). Rather, the sample size was based on Julious[20] who suggests a group size of at least 12 to obtain reasonably precise estimates of an effect and its variance.

### Participant flow

During the 15-month recruitment period, potential participants will attend an initial ear, nose and throat (ENT consultation with the local principal investigator (PI). If they are deemed a potential candidate for the study, the PI will ask the patient for verbal consent to audiorecord subsequent conversations where information about the VOCALIST study is given. The PI will then provide the participant information sheet (PIS). After the consultation, the local research nurse (RN) will go through the PIS with the patient and answer any questions the patient

may have. This conversation will also be audiorecorded and during this conversation the RN will ask the patient to complete written informed preconsent for the conversations to be audiorecorded and analysed by the qualitative research team.

If the patient is willing to participate in the study, the RN will arrange to call the patient within 1–3 days to book a date for the screening appointment. During the telephone call (also audiorecorded) the RN will answer any further patient questions and give the patient the opportunity to change their mind about study participation if they wish. If the patient confirms they would like to participate, the RN will book the patient's screening appointment at their local site.

At the screening and baseline appointments, informed consent will be obtained by the RN prior to the assessments being performed. Throughout the recruitment process the RN will remind patients that they are under no obligation to enter the study and that they can withdraw at any time, without having to give a reason. No clinical trial procedures will be conducted prior to taking consent from the participant. Consent will not denote enrolment into trial. Figure 1 shows the flow of patients through the study and the assessments performed at each visit.

### Randomisation

Participants will be allocated to either of the two operations (see figure 1) using block randomisation stratified by centre to ensure that approximately equal numbers of patients are allocated to each treatment arm within each centre. The randomisation will be carried out using the web-based service from the company 'Sealed Envelope'. Varying block sizes will be used and these will only be known to the statistician and Sealed Envelope.

### Data analysis plan

#### Quantitative data analysis

Descriptive analyses (means, SD and proportions) will be used to summarise the characteristics of the patients recruited to the study. A consort flow diagram will be produced to show the progress of the participants through the phases of the feasibility study and describe the follow-up at the various time-points. The study dataset will be analysed on an intention-to-treat basis. The primary and secondary outcomes will be analysed descriptively (using either proportions, or means, as appropriate) for each group. Differences between groups will be estimated and presented with 95% CIs. No formal comparisons will be made as the study is not powered for this. Stata V.14 will be used for analysis.

#### Qualitative data analysis

Patient eligibility and recruitment pathway details will be mapped for each centre to include: the point at which patients receive information about the trial and encounter members of the clinical team and the timing and frequency of appointments. Logs of eligible and recruited patients will be assembled using simple flow

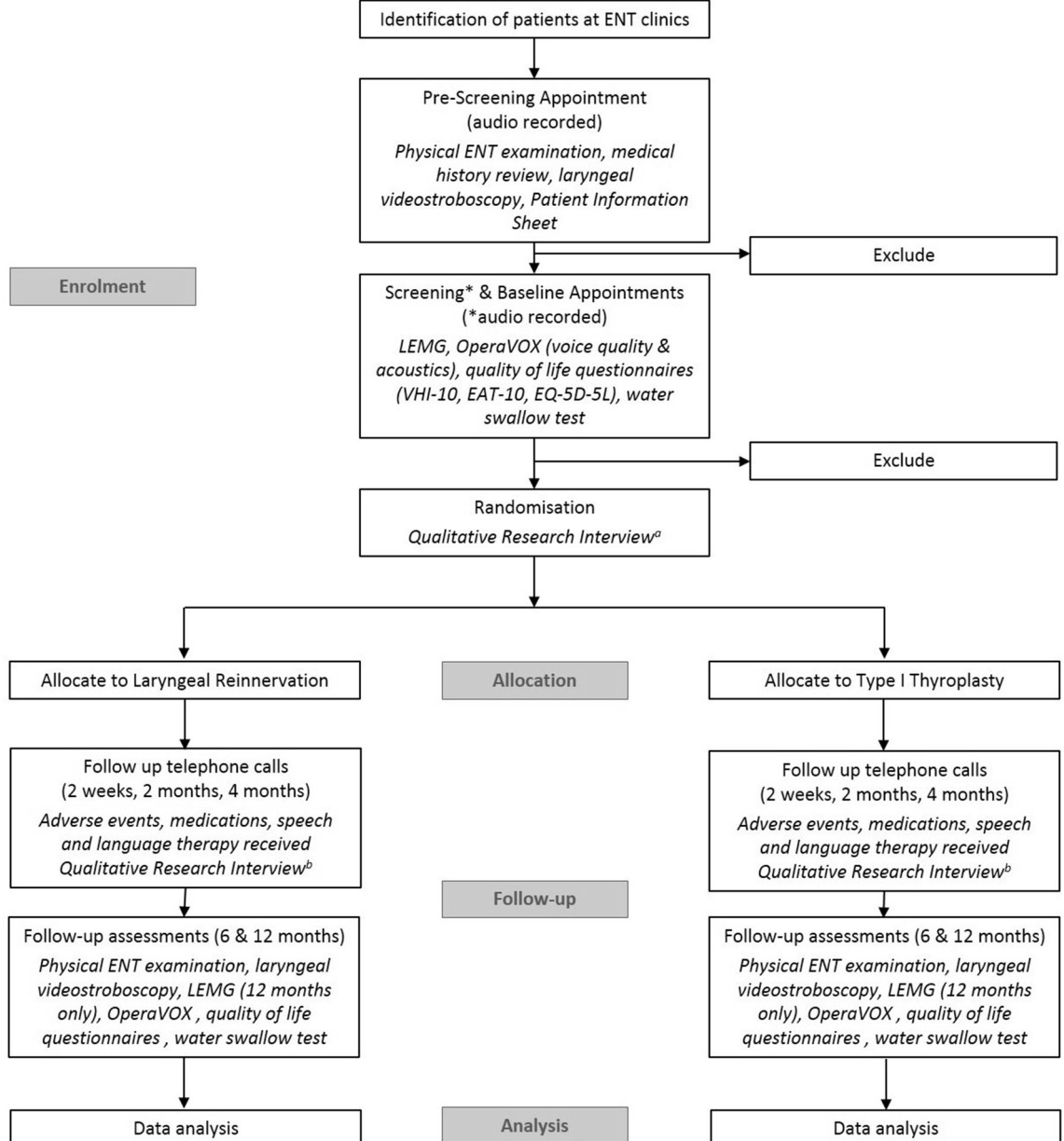

Figure 1 Flow diagram of the VOCALIST feasibility study. EAT-10, Eating Assessment Tool 10-items; ENT, ear, nose and throat; EQ-5D-5L; EuroQoL 5 Dimensions 5 Levels; LEMG, laryngeal electromyography; OperaVOX, On person Rapid Voice Examiner; VHI-10, Voice Handicap Index 10.

charts and counts to display numbers and percentages of patients at each stage of the eligibility and recruitment process. Audio recordings of the recruitment conversations will be analysed using established techniques of applied conversation analysis[23] to examine how patient consultations proceed and to identify any instances of interactional trouble which may impact on recruitment.

In-depth interviews will be conducted by the Qualitative Researcher with Trial staff (n=10 approximately, including members of the SMG, the CI and those most involved in the design and management of the trial and with clinical and recruitment staff across the three

centres) and participants eligible for recruitment to the study (n=10 approximately, including those who accept and decline randomisation). Interviews will be analysed thematically, using a constant comparison methodology until a point where theoretical saturation is accomplished.[24]

### Data management
Quantitative data will be collected at the trial sites using paper CRFs and entered centrally by the Trial Manager onto the online study database (provided by Sealed Envelope). Any local data collection or data entry queries

raised by local sites will be brought to the attention of the central London team (TM). Identification logs, screening logs and enrolment logs will be kept at the trial site in a locked cabinet within a secured room.

Qualitative data will be recorded on encrypted audio recorders and transferred to the University of Bristol's secure Research Data Storage Facility (RDSF). Interviews will be transcribed verbatim, with personal identifiers removed. Files will be stored on the University of Bristol's secure RDSF.

At the end of the study, all study-related documents will be archived. The chief investigator will archive the study master file for 15 years in line with all relevant legal and statutory requirements. The principal investigator at each participating site will archive his/her respective site's study documents for 15 years in line with all relevant legal and statutory requirements.

### Study monitoring

The study will be managed by the Study Management Group (SMG) consisting of the coinvestigators, site PIs and patient representative. They will meet every 2 months during the set up and duration of the study.

The study will be overseen by the Study Steering Committee (SSC), consisting of an independent chair, an independent clinician, an independent speech and language therapist and a patient representative. They will meet biannually with the CI and relevant members of the SMG.

### Patient and public involvement

This research has been developed in response to patient demand for a trial which will deliver the answers they need to make an informed choice regarding their care. Patients and the public have been involved in confirming the need for the research, developing the trial design, choice of outcome measure and wording of patient documentation. Two patient representatives are active members of the SMG and SSC and have been provided with lay research training and support to ensure they understand their role in the research project, remain engaged as active partners and feel confident in their relationships with the research team. These patient representatives will continue to be involved throughout the feasibility study, including assisting with optimising recruitment and retention, interpreting the findings and disseminating the results to their formal and informal patient networks.

### Ethics and dissemination

A favourable ethical approval was granted by the NRES—Committee Bromley (reference number 11/LO/0583) prior to commencement of the study. Local approvals for each site have been sought prior to commencement of recruitment. All members of the research team and at each site will be trained in the aspects of good clinical practice appropriate to their role in the study.

To ensure the results from this feasibility study can be taken forward into a plan for a future large-scale RCT, there is a need to engage all those involved in voice health including ENT surgeons, speech therapists, voice and singing teachers, general practitioners and practice nurses. Therefore, results will be disseminated through the publication of articles in peer-reviewed medical journals targeted at health professionals involved in the care of patients with voice disorders and presented at national and international scientific meetings. Our findings and future trial plans will be discussed with relevant professional organisations including the British Laryngological Association (BLA) and Royal Colleges to ensure they are disseminated to their members through electronic newsletters, websites and national meetings. Our Patient and Public Involvement representatives will assist in disseminating results in a suitable lay language to relevant journals and patient information sections of health-related websites, ensuring that patients and the public are both informed and engaged for the future RCT.

### Study status

At the time of manuscript submission, the feasibility study is open to recruitment.

### CONCLUSION

Reports on short and long-term voice outcomes of patients who undergo thyroplasty are sparse and methodologically poor. The quality of the current literature regarding laryngeal reinnervation is also poor. For these reasons, the experienced multidisciplinary VOCALIST research team have designed this feasibility study to provide reliable information on how to conduct a future multicentre trial to determine the clinical and cost-effectiveness of laryngeal reinnervation versus thyroplasty for patients with UVFP. The combination of quantitative and qualitative approaches and the support of the BLA and National Institute for Health Research Clinical Research Network will ensure that all aspects of feasibility are assessed, including patient recruitment strategies and appropriate outcome measures. The lessons learnt in this feasibility study will ensure the success of a future RCT (including whether training courses need to be developed to enable greater numbers of UK surgeons to perform laryngeal reinnervation surgery) ultimately enabling an evidence-based pathway of care to be developed for patients with UVFP.

**Author affiliations**
[1]Ear Institute, University College London, London, UK
[2]School of Allied Health, Australian National Catholic University, North Sydney, New South Wales, Australia
[3]School of Social and Community Medicine, University of Bristol, Bristol, UK
[4]Faculty of Medicine, National University of Malaysia, Malaysia
[5]Statistical Science, University College London, London, UK
[6]Department of Applied Health Research, University College London, London, UK
[7]UCL Hospitals Biomedical Research Centre, University College London Hospitals NHS Foundation Trust, London, UK

[8]Royal National Throat, Nose and Ear Hospital, University College London Hospitals NHS Foundation Trust, London, UK

[9]Department of Otolaryngology—Head & Neck Surgery, Manchester Royal Infirmary, Manchester, UK

[10]Department of Otolaryngology, Poole Hospital, Poole, UK

**Acknowledgements** The authors would like to thank Eilis Longley-Brown and Jaqueline Sayers for their dedicated work as patient representatives on the VOCALIST SMG and SSC, respectively. The authors would also like to thank the Chair, Mr Julian McGlashan, and members of the SSC as well as the site study staff and VOCALIST participants.

**Contributors** MB is the chief investigator and led the study design. HB coordinated the grant acquisition, study design, protocol development and drafted the manuscript. PC, MJ, GA, MMB, SM and AS contributed to study design, grant acquisition and protocol development. RY and AD designed and conducted the Patient and Public Involvement work. YK, KH, SH, HK and CW contributed to protocol development. All authors have revised the manuscript and approved the final version.

**Funding** This paper presents independent research funded by the National Institute for Health Research (NIHR) under its Research for Patient Benefit (RfPB) Programme (Grant Reference Number PB-PG-1013-32058). The views expressed are those of the authors and not necessarily those of the NHS, the NIHR or the Department of Health. This manuscript relates to version 4 of the study protocol, dated 03 March 2017.

**Competing interests** None declared.

**Ethics approval** NRES–Committee Bromley (11/LO/0583).

**Provenance and peer review** Not commissioned; externally peer reviewed.

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
