## [Reviewer comments · BMJ Open]

ARTICLE DETAILS

TITLE (PROVISIONAL)	Does Laryngeal Reinnervation or Type I Thyroplasty give better voice results for patients with Unilateral Vocal Fold Paralysis (VOCALIST): study protocol for a feasibility randomised controlled trial
AUTHORS	Blackshaw, Helen; Carding, Paul; Jepson, Marcus; Mat Baki, Marina; Ambler, Gareth; Schilder, Anne; Morris, Stephen; Degun, Aneeka; Yu, Rosamund; Husbands, Samantha; Knowles, Helen; Walton, Chloe; Karagama, Yakubu; Heathcote, Kate; Birchall, Martin

VERSION 1 - REVIEW

REVIEWER	Ass Prof Rajeev Parameswaran National University Hospital Lower Kent Ridge Road Singapore 119074
REVIEW RETURNED	07-May-2017

GENERAL COMMENTS	This study is a feasibility study to look into running a RCT between 2 approaches in the management of UFVP. This study should be welcomed to address an important area of clinical practice that is less well treated and managed. The study is well designed and looks to overcome problems generally faced in a RCT.
---

REVIEWER	Randal Paniello Washington University, St. Louis USA
REVIEW RETURNED	08-May-2017

GENERAL COMMENTS	I disagree with the statement in the abstract that "there is no robust evidence..." but would change this to "there is limited evidence..." The randomized clinical trial that was "attempted" in the US was actually completed and published, although with fewer patients than planned as the manuscript states. nevertheless, it did provide some useful data that answers some of the feasibility questions proposed herein, and also some evidence about the best option for adults. The abstract also implies that there will be one best option for all adults with UVFP determined from the planned study. This is unlikely; it is more likely that certain subgroups of adults will benefit from one approach vs another. In our randomized trial, we found that in younger patients (< age 52), reinnervation was better, but in older patient thyroplasty was better. We also feel that patients with limited life expectancy (e.g., lung cancer) with UVFP are not best served by reinnervation due to the time needed for the operation to take effect. So, the premise that there will be one operation that "wins" for all
--

adults, is flawed.

The introduction states that silastic type I thyroplasty is the only permanent surgical option other than reinnervation. This omits other types of thyroplasty (e.g., Gore-tex implants, which have become the most commonly used implant materials in the US) as well as other types of surgical medialization (such as arytenoid adduction).

The short follow-up period of 6 months is a bit of a problem. Our experience has been that most, but not all, patients undergoing reinnervation begin to experience benefit at about 4 months, but some report continued improvement that occurs beyond 6 months; meanwhile, thyroplasty patients usually plateau around 1-2 months post-op, and some have a diminished result at 6-12 months due to atrophy that occurs from prolonged denervation. The data from our randomized trial, in which we collected data at 6 and 12 months post-op, reflect this. By reporting only 6 month data, the study will be biased in favor of thyroplasty. But many of the data items are to be collected at 12 months as well, so the statement that the study is limited to 6 months for budget reasons is a little confusing.

I see cricoarytenoid joint fixation as a potentially problematic exclusion criterion, because 1) it is difficult to diagnose, even with direct laryngoscopy under general anesthesia which most patients do not undergo; 2) it can occur subsequent to a reinnervation procedure, prior to the outcome measures; 3) it is not a contraindication to thyroplasty; 4) patients with CA joint fixation can still benefit from reinnervation as well, as tone is reestablished. Fortunately this is rare, but I would take it out of the exclusion criteria.

The feasibility study aims to determine whether a large-scale study, ostensibly to provide the definitive answer to the primary question, can be carried out in the UK. To do this, some idea of the larger sample size needed should be addressed. If the pilot study is 30 patients, how many are needed for the "big" study? Will it need to be carried out at more than 3 sites? If so, can additional surgeons be brought up to speed on reinnervation? It would seem these questions should also be addressed; otherwise, we could find that the big study is feasible from the perspectives of the pilot study and yet unfeasible for lack of study sites or surgeons.

In addition, some statement should be added that indicates the number of patients the PI is currently seeing that would qualify for the feasibility study, i.e., how long is it expected to take to accrue the 30 patients planned?

We determined that the use of post-op speech therapy was a confounder that we could not control to our satisfaction; the quality of therapy, number of sessions (which could be zero), etc. could significantly affect the results. The only way to equalize this was to require that no patient had post-op speech therapy until the final data collection was obtained at 12 months. I do not see this important confounder addressed in the study plan. Maybe it is not as important for the pilot study, but the feasibility of declining SpTx in the post-op period may be of interest for planning the big study.

One of the feasibility goals is to determine whether the study can be presented to patients with equipoise. It is not stated whether the

	investigators themselves do, in fact, have clinical equipoise at the outset of the study; this is important. We found that some participants in our multicenter trial did not have this and were thus unable to recruit effectively. It is also unclear how this goal will be determined, i.e., how will the presence of equipoise be determined for the pilot study? The addition of swallowing measures is interesting and important, but adds cost and should be considered carefully for a study that is aimed at determining the best operation for voice restoration. Patients with particularly wide posterior glottal gaps are the ones most at risk for swallowing problems; our experience has been that reinnervation does not close a wide gap very well. Therefore we do not offer reinnervation to patients with high vagal lesions (e.g., from skull base surgery at the jugular foramen) who have combined RLN and SLN deficits and wider glottal gaps. The study might consider excluding these patients, in which case the swallowing aspects could probably be deleted and cost saved. In the reference list, I was surprised to see our multicenter trial paper not included. (Not a major concern though.) Otherwise, the study seems to be designed quite well and is poised to answer the research questions well. The PI and the research team have done a nice job of preparing for the study, and they should be congratulated. It will be very interesting to see how it turns out. Finally, I would be willing to help with this study if some sort of outside reviewer or monitor is needed, if someone from the US can be used in this role.
--	---

REVIEWER	David O. Francis Vanderbilt University Medical Center USA
REVIEW RETURNED	15-May-2017

GENERAL COMMENTS	Comments It is my pleasure to review this research protocol. I found it very well considered and timely. The question to be answered is important. The optimal treatment for persistent UVFP is not known. Overall, the authors should be applauded for putting together this protocol. However, I have several concerns that I would like to see addressed and or responded to. These are listed below. Thank you for the opportunity to review this protocol and please let me know if I can be of future assistance.  1. One of the biggest concerns is the idea of equipoise. Do the authors feel that the similar UVFP patients are truly candidates for either procedure to address their disability? Is this the practice of the participating surgeons/centers?
--

2. Significant variability exists in how Type I laryngoplasty is performed. It appears that the authors intend to use prefabricated silastic implants. This is reasonable. However, it should be known and comment provided that many different techniques and methodologies exist (handcarved implants, gortex, among others). Furthermore, in many surgeons hands, prefabricated implants do not or may not yield the best outcomes. Similarly, many surgeons may decide pre- or intraoperatively to do arytenoid procedures to optimize laryngoplasty results. This is not discussed herein. Please be clear why you are defining type I laryngoplasty using the method discussed.
3. Type I laryngoplasty and reinnervation procedures are highly operator-dependent. Volume of procedures performed matters. It is important the surgeons who may be performing these procedures have similar/adequate experience doing these procedures and do them using a similar approach. If there is a broad range in experience among surgeons, it could affect results. Please comment on experience of the 3 centers doing the procedures in terms of their volume and outcomes. Do surgeons at these centers currently use these procedures interchangeably?
4. Follow-up – outcomes need to be followed greater than 6 months following the intervention. This is particularly true for reinnervation, but also for laryngoplasty. I would recommend 1 year of follow-up for symptoms to stabilize.

Introduction

- It is questionable whether many surgeons agree that type 1 laryngoplasty routinely takes 1 hour.
- Also, note (as above) that there are many different techniques and variations on laryngoplasty, which are not discussed. Please take care to be explicit what is meant by thyroplasty and, if variations are allowed (e.g., arytenoid procedures), then put into protocol these allowances.
- Paragraph 2 – Despite lack of sufficient data, authors make some statements about the effectiveness of reinnervation that have not been well justified in prior studies. For example, “As such, it reestablishes the tone and bulk...” This is the intent of the procedure, but it has not been consistently and definitively shown that this is the reliable outcome. It might be better to temper statements like this within this paragraph using the modifier “may” instead of definitive statements on its effectiveness.

Methods

Authors provide an excellent description of the qualitative portion. This is of critical importance and is well done.

Study design:

1. Single blinding = need to comment/ensure that the incisions used for both procedures are similar in size so that the blinded reviewers are not able to differentiate who had what

	procedure. Participants:  1. The first inclusion criterion indicates that patients may have had the UVFP between 6 and 60 months. This is a broad range. Patients whose condition has been present for 6 months may have further spontaneous recovery, whereas those who have had the condition for 60 months would be unlikely to have any additional physiological or symptomatic improvement. 2. Why used a VHI-10 of ≥ 16? I am not sure this is a validated cut-off. Please explain. 3. GRBAS – many have argued that CAPE-V is a better perceptual tool. Please explain why you have chosen GRBAS instead. Interventions  1. As mentioned previously, I would recommend the authors expand on what is allowed with type I laryngoplasty. As it stands, the authors describe only the use of prefabricated silastic implants without any arytenoid procedures. Are surgeons allowed to use different materials or to do additional arytenoid procedures if they feel that it will improve outcome? Please explain. Outcomes  2. Please explain the use of VHI-10 and its validation for use in this patient population. 3. GRBAS – please explain rationale for its use over CAPE-V 4. Vocal fold vibration (secondary outcome 4): “Data will be collected at baseline, 6 and 23 months...” I was not aware of this collection time. It may be helpful to put a diagram or table showing what outcomes are being collected at different time points beyond that shown in the flow diagram provided.
--	--

VERSION 1 – AUTHOR RESPONSE

Reviewer: 1

Ass Prof Rajeev Parameswaran
Institution and Country
National University Hospital
Lower Kent Ridge Road
Singapore
119074

This study is a feasibility study to look into running a RCT between 2 approaches in the management of UFVP. This study should be welcomed to address an important area of clinical practice that is less

well treated and managed. The study is well designed and looks to overcome problems generally faced in a RCT.

Thank you for your positive comments.

Reviewer: 2

Randal Paniello
Institution and Country
Washington University, St. Louis
USA

I disagree with the statement in the abstract that "there is no robust evidence..." but would change this to "there is limited evidence..."

We agree and have amended the manuscript accordingly (page 2).

The randomized clinical trial that was "attempted" in the US was actually completed and published, although with fewer patients than planned as the manuscript states. nevertheless, it did provide some useful data that answers some of the feasibility questions proposed herein, and also some evidence about the best option for adults. The abstract also implies that there will be one best option for all adults with UVFP determined from the planned study. This is unlikely; it is more likely that certain subgroups of adults will benefit from one approach vs another. In our randomized trial, we found that in younger patients (< age 52), reinnervation was better, but in older patient thyroplasty was better. We also feel that patients with limited life expectancy (e.g., lung cancer) with UVFP are not best served by reinnervation due to the time needed for the operation to take effect. So, the premise that there will be one operation that "wins" for all adults, is flawed. Given the difficulties encountered in terms of recruitment to the US study.

We agree and have amended the language of our abstract accordingly (page 2).

The introduction states that silastic type I thyroplasty is the only permanent surgical option other than reinnervation. This omits other types of thyroplasty (e.g., Gore-tex implants, which have become the most commonly used implant materials in the US) as well as other types of surgical medialization (such as arytenoid adduction).

We have amended our abstract to better explain the surgical options available (page 2).

The short follow-up period of 6 months is a bit of a problem. Our experience has been that most, but not all, patients undergoing reinnervation begin to experience benefit at about 4 months, but some report continued improvement that occurs beyond 6 months; meanwhile, thyroplasty patients usually plateau around 1-2 months post-op, and some have a diminished result at 6-12 months due to atrophy that occurs from prolonged denervation. The data from our randomized trial, in which we collected data at 6 and 12 months post-op, reflect this. By reporting only 6 month data, the study will be biased in favor of thyroplasty. But many of the data items are to be collected at 12 months as well, so the statement that the study is limited to 6 months for budget reasons is a little confusing.

We have amended the wording on page 3 to make our manuscript clearer in this regard. We agree that collecting data at 12 months as well as 6 months is important and are aiming to follow-up as many patients as possible for the full 12-month follow-up period. However, due to our funding being granted primarily on the basis of exploring the feasibility of recruiting patients to this study, we do not

potentially have the resources available to follow up all patients for 12 months. Those recruited towards the end of the study will therefore be followed up for 6 months.

I see cricoarytenoid joint fixation as a potentially problematic exclusion criterion, because 1) it is difficult to diagnose, even with direct laryngoscopy under general anesthesia which most patients do not undergo; 2) it can occur subsequent to a reinnervation procedure, prior to the outcome measures; 3) it is not a contraindication to thyroplasty; 4) patients with CA joint fixation can still benefit from reinnervation as well, as tone is reestablished. Fortunately this is rare, but I would take it out of the exclusion criteria.

Thank you for the feedback, we will take this into account for the future RCT design.

The feasibility study aims to determine whether a large-scale study, ostensibly to provide the definitive answer to the primary question, can be carried out in the UK. To do this, some idea of the larger sample size needed should be addressed. If the pilot study is 30 patients, how many are needed for the "big" study? Will it need to be carried out at more than 3 sites? If so, can additional surgeons be brought up to speed on reinnervation? It would seem these questions should also be addressed; otherwise, we could find that the big study is feasible from the perspectives of the pilot study and yet unfeasible for lack of study sites or surgeons.

We completely agree and will certainly be taking these points into account when analysing the results from this feasibility study and planning our larger RCT.

The sample size for the future (definitive) study will depend on our choice of primary outcome and the anticipated 'effect size' (i.e. the expected difference in effects of the two techniques). Both of these will be somewhat informed by the results of this feasibility study and hence we are currently unable to derive an approximate sample size for the future study.

The recruitment rates observed at the 3 sites during this feasibility study will enable us to plan the number of sites required to recruit the larger sample size.

Currently there are only 4 surgeons in the UK routinely carrying out reinnervation surgery with many others undergoing the required training. Ahead of the larger RCT, our research team are working to ensure we will have sufficient numbers of suitably experienced surgeons at the required number of sites through a process of checking they have attended accredited training courses and completed a required number of operations both under supervision and on their own.

In addition, some statement should be added that indicates the number of patients the PI is currently seeing that would qualify for the feasibility study, i.e., how long is it expected to take to accrue the 30 patients planned?

A sentence has been added into the manuscript to reflect the planned 15-month recruitment period (page 8).

We determined that the use of post-op speech therapy was a confounder that we could not control to our satisfaction; the quality of therapy, number of sessions (which could be zero), etc. could significantly affect the results. The only way to equalize this was to require that no patient had post-op speech therapy until the final data collection was obtained at 12 months. I do not see this important confounder addressed in the study plan. Maybe it is not as important for the pilot study, but the feasibility of declining SpTx in the post-op period may be of interest for planning the big study.

At this stage, whilst focusing on feasibility of recruitment, we made the decision to allow speech and language therapy (SLT) to continue as per normal clinical practice.

We completely agree that SLT needs to be controlled post-operatively when assessing efficacy of the two operations in the larger RCT. As such, we are in the process of conducting separate research

(PhD project) into the current forms of SLT used for UVFP, so sessions can be appropriately planned for the larger RCT protocol. This research will also be published at the end of this feasibility study.

One of the feasibility goals is to determine whether the study can be presented to patients with equipoise. It is not stated whether the investigators themselves do, in fact, have clinical equipoise at the outset of the study; this is important. We found that some participants in our multicenter trial did not have this and were thus unable to recruit effectively. It is also unclear how this goal will be determined, i.e., how will the presence of equipoise be determined for the pilot study?

We have involved all of our PIs from the start of this feasibility study design to ensure they understand the purpose of this study and they have stated that they are in equipoise at the start. All three PIs have contributed to the protocol design and attend our regular study research meetings.

The qualitative component of the study explores whether, in practice, study information is presented to patients in a way that demonstrates clinicians' equipoise. This is achieved through an analysis of a recorded consultation data, examining whether the two treatment options are described, and understood by patients, as being of equal benefit.

The addition of swallowing measures is interesting and important, but adds cost and should be considered carefully for a study that is aimed at determining the best operation for voice restoration. Patients with particularly wide posterior glottal gaps are the ones most at risk for swallowing problems; our experience has been that reinnervation does not close a wide gap very well. Therefore we do not offer reinnervation to patients with high vagal lesions (e.g., from skull base surgery at the jugular foramen) who have combined RLN and SLN deficits and wider glottal gaps. The study might consider excluding these patients, in which case the swallowing aspects could probably be deleted and cost saved.

At this feasibility stage, we set out to collect swallowing-related measurements to allow us to provide further answers to this hypothesis and contribute to our planning for the design of the larger future trial.

In the reference list, I was surprised to see our multicenter trial paper not included. (Not a major concern though.)

Since we refer to your trial we agree this should have been referenced and so we have amended the manuscript accordingly (page 4).

Otherwise, the study seems to be designed quite well and is poised to answer the research questions well. The PI and the research team have done a nice job of preparing for the study, and they should be congratulated. It will be very interesting to see how it turns out.

Finally, I would be willing to help with this study if some sort of outside reviewer or monitor is needed, if someone from the US can be used in this role.

Thank you for your thorough review and positive comments. Your offer of international review is most appreciated and we look forward to contacting you in the future in this regard.

Reviewer: 3

David O. Francis
Institution and Country
Vanderbilt University Medical Center
USA

It is my pleasure to review this research protocol. I found it very well considered and timely. The question to be answered is important. The optimal treatment for persistent UVFP is not known. Overall, the authors should be applauded for putting together this protocol. However, I have several concerns that I would like to see addressed and or responded to. These are listed below. Thank you for the opportunity to review this protocol and please let me know if I can be of future assistance.

Thank you for your thorough review and positive feedback.

1. One of the biggest concerns is the idea of equipoise. Do the authors feel that the similar UVFP patients are truly candidates for either procedure to address their disability? Is this the practice of the participating surgeons/centers?

The surgeons in this study have stated from the outset that they are open to patients receiving either operation in their clinical practice and that they do not have a personal preference to offer one operation over the other, due to the lack of existing evidence. Our qualitative work establishes whether this equipoise is present during consultations with VOCALIST patients and if the two operations are presented in a balanced manner. Where they are not, the qualitative researchers will provide guidance to recruiters in how to be more balanced in information delivery.

2. Significant variability exists in how Type I laryngoplasty is performed. It appears that the authors intend to use prefabricated silastic implants. This is reasonable. However, it should be known and comment provided that many different techniques and methodologies exist (handcarved implants, gortex, among others). Furthermore, in many surgeons hands, prefabricated implants do not or may not yield the best outcomes. Similarly, many surgeons may decide pre- or intraoperatively to do arytenoid procedures to optimize laryngoplasty results. This is not discussed herein. Please be clear why you are defining type I laryngoplasty using the method discussed.

Thank you for your suggestions. We have amended the abstract to contain further details on the different techniques available for surgical medialisation (page 2). We have also amended the intervention section regarding concurrently permitted surgery and cutting of the silastic block into the correct size for the patient (page 6).

3. Type I laryngoplasty and reinnervation procedures are highly operator dependent. Volume of procedures performed matters. It is important the surgeons who may be performing these procedures have similar/adequate experience doing these procedures and do them using a similar approach. If there is a broad range in experience among surgeons, it could affect results. Please comment on experience of the 3 centers doing the procedures in terms of their volume and outcomes. Do surgeons at these centers currently use these procedures interchangeably?

The 3 surgeons in this study offer both operations in their clinical practice. They are experienced surgeons who have conducted in excess of 10 operations of each type. At the protocol-writing stage of this study they met to agree on the methods to be used for each operation so that the same techniques are used across the 3 sites. These details are included within the main study protocol, copies of which are held at each site.

4. Follow-up – outcomes need to be followed greater than 6 months following the intervention. This is particularly true for reinnervation, but also for laryngoplasty. I would recommend 1 year of follow-up for symptoms to stabilize.

We completely agree and are collecting data at 12 months where possible (funding constraints may limit some of the latterly-recruited patients to 6 months follow up). We have amended the manuscript

accordingly (page 3).

Introduction

- It is questionable whether many surgeons agree that type 1 laryngoplasty routinely takes 1 hour.

We agree that surgery times are both surgeon- and patient-dependant, however in our group of 3 surgeons the median time for this operation is around the 1 hour mark.

- Also, note (as above) that there are many different techniques and variations on laryngoplasty, which are not discussed. Please take care to be explicit what is meant by thyroplasty and, if variations are allowed (e.g., arytenoid procedures), then put into protocol these allowances.

We agree and the introduction sections now contains relevant detail (page 4).

- Paragraph 2 – Despite lack of sufficient data, authors make some statements about the effectiveness of reinnervation that have not been well justified in prior studies. For example, “As such, it reestablishes the tone and bulk...” This is the intent of the procedure, but it has not been consistently and definitively shown that this is the reliable outcome. It might be better to temper statements like this within this paragraph using the modifier “may” instead of definitive statements on its effectiveness.

We have modified the manuscript language accordingly (page 4) and included LEMG measurements within our protocol to establish that post-operatively the reinnervation observed is from the surgical procedure and not spontaneous recovery.

Methods

Authors provide an excellent description of the qualitative portion. This is of critical importance and is well done.

Our qualitative researchers would like to thank you for the positive comments.

Study design:

1. Single blinding = need to comment/ensure that the incisions used for both procedures are similar in size so that the blinded reviewers are not able to differentiate who had what procedure.

The blinded reviewers are the neurophysiologists assessing the LEMG data and the speech and language therapists assessing the acoustic data. As such, none of the reviewers will physically see the patients and instead will only be presented with anonymised recorded data for analysis.

Participants:

1. The first inclusion criterion indicates that patients may have had the UVFP between 6 and 60 months. This is a broad range. Patients whose condition has been present for 6 months may have further spontaneous recovery, whereas those who have had the condition for 60 months would be unlikely to have any additional physiological or symptomatic improvement.

Our feasibility study only aims to include patients once they are considered clinically to be candidates for surgery. We kept the age range broad here so that it could be taken into context with the patient's

scenario as well as the other inclusion criteria. For example, patients with obvious nerve damage that would not spontaneously recover, could then be included in the study. Our LEMG analysis at the screening stage assists us to define those patients not likely to spontaneously recover.

2. Why used a VHI-10 of ≥ 16 ? I am not sure this is a validated cut-off. Please explain.

Vocal Handicap-10 scores range from 0-40. Arffa et al (2012) suggested that a VHI-10 score of >11 should be considered abnormal. For pragmatic reasons we have chosen VHI cut-off score of >16 (i.e. moderate voice vocal handicap) in order to be able to detect change. There is no sensitivity to change data available. The validation of this cut-off will be examined as part of the feasibility evaluation.

3. GRBAS – many have argued that CAPE-V is a better perceptual tool. Please explain why you have chosen GRBAS instead.

The GRBAS scale was chosen because it is supported by robust reliability and validity evidence (De Bodt et al 1997, Nemr et al 2012) and is extensively used in the international literature.

Interventions

1. As mentioned previously, I would recommend the authors expand on what is allowed with type I laryngoplasty. As it stands, the authors describe only the use of prefabricated silastic implants without any arytenoid procedures. Are surgeons allowed to use different materials or to do additional arytenoid procedures if they feel that it will improve outcome? Please explain.

The manuscript detail has been amended accordingly (page 6).

Outcomes

2. Please explain the use of VHI-10 and its validation for use in this patient population.

The Vocal Handicap Index was chosen because it is extensively used in the international literature. The VHI-10 has been reported to be highly correlated with VHI-30 and takes less time for the patient to complete without loss of validity (Rosen et al 2004). The VHI-10 has been translated and validated in many languages.

3. GRBAS – please explain rationale for its use over CAPE-V

Please see comments above.

4. Vocal fold vibration (secondary outcome 4): “Data will be collected at baseline, 6 and 23 months...” I was not aware of this collection time. It may be helpful to put a diagram or table showing what outcomes are being collected at different time points beyond that shown in the flow diagram provided.

We apologise for the typing error, this should read “baseline, 6 and 12 months” in line with the flow chart. We have amended this in the manuscript accordingly (page 7).